

# Development and validation of a simple clinical nomogram for predicting infectious diseases in pediatric kidney transplantation recipients: a retrospective study

Li Li[1,2,*], Meng Fu[1,*], Changshan Wang[2], Yuxin Pei[1], Lizhi Chen[1], Liping Rong[1], Yuanyuan Xu[1], Zhilang Lin[1], Yuanquan Qiu[1], Xiaoyun Jiang[1] and Mengjie Jiang[1]

[1] Department of Pediatric Nephrology and Rheumatology, The First Affiliated Hospital of Sun Yat-sen University, Guangzhou, Guangdong, China
[2] Shenzhen Guangming District People's Hospital, Shenzhen, Guangdong, China
[*] These authors contributed equally to this work.

Corresponding authors
Xiaoyun Jiang,
jxiaoy@mail.sysu.edu.cn
Mengjie Jiang,
jiangmj8@mail.sysu.edu.cn

## ABSTRACT

To construct and verify an easy-to-use nomogram for predicting the risk of infectious diseases in pediatric kidney transplant recipients. Clinical data of hospitalized pediatric kidney transplant recipients were retrospectively analyzed. Meaningful variables identified from the multivariate stepwise logistic regression analysis were used to construct the nomogram. Internal validation was performed using Bootstrap resampling 1,000 times. The nomogram was evaluated using calibration, decision and receiver operating characteristic (ROC) curves. A total of 297 pediatric kidney transplant recipients were included (164 infected, 133 non-infected). Multivariate stepwise regression analysis identified white blood cell count (WBC), lymphocyte to monocyte ratio (MLR), platelet to neutrophil ratio (PNR), red cell distribution width-standard deviation (RDW-SD), and albumin (ALB) as significant predictors of postoperative infection. The nomogram, based on the five indicators, showed strong discrimination ability (AUC = 0.756; 95% CI [0.702–0.811]), with a sensitivity of 88.0% and a specificity of 54.3%. The calibration curve and decision curve further demonstrated good consistency and clinical practicality between the predicted and actual values. WBC, MLR, PNR, RDW-SD, and ALB are effective indicators for predicting postoperative infection in pediatric kidney transplant recipients. The nomogram constructed from these indicators can effectively predict and evaluate the early risk of infection in these patients.

# INTRODUCTION

In 1954, the world's first kidney transplant was performed in the United States (*Tan & Merchant, 2019*). In recent years, kidney transplantation technology has advanced rapidly and is now considered the most effective treatment for end-stage kidney disease (*Saran*

*et al., 2019*). The immune system of kidney transplant recipients is suppressed due to the use of multiple immunosuppressants following surgery, resulting in a higher risk of infection compared to the general population. This risk is particularly significant among pediatric transplant recipients (*Knackstedt & Danziger-Isakov, 2017*; *Scaggs Huang & Danziger-Isakov, 2019*).

Infection is a critical factor influencing the postoperative survival of kidney transplant recipients (*Aguado et al., 2018*). It ranks among the most frequent complications following transplantation and is a significant contributor to kidney transplant failure (*Dharnidharka et al., 2006*; *Dharnidharka, Stablein & Harmon, 2004*). Therefore, the early and prompt identification of post-transplantation infection holds significant clinical research value.

Pediatric kidney transplant recipients represent a distinct group with heightened susceptibility to infections compared to adults (*Hebert et al., 2017*). Studies indicate that infections and gastrointestinal problems are the most common reasons for emergency visits among pediatric kidney transplant recipients (*King et al., 2022*). The risk factors for infections vary by type, including younger age at transplantation, use of immunosuppressants, ureteral epithelial damage, occurrence of acute rejection, and gender (*Arpali et al., 2020*; *Avcı et al., 2022*; *Levi et al., 2022*; *McCaffrey, Bhute & Shenoy, 2021*). While some clinical studies have examined infection following kidney transplantation in adults (*Zhang et al., 2024*), there remains a notable gap in clinical prediction tools tailored specifically for pediatric populations. This study aims to address this gap by developing a predictive model to assess infection risk in pediatric kidney transplant recipients, thereby enhancing clinical decision-making.

Nomograms aid in clinical decision-making by integrating and assigning scores to variables to predict the probability of clinical events (*Jiang et al., 2022*; *Wang et al., 2022*). During follow-up visits for pediatric kidney transplant recipients, routine examinations typically include a complete blood count (CBC) as well as liver and kidney function tests. Among these, the CBC stands out as the most basic, routine, and readily available test.

Therefore, this study aims to develop and validate a clinical nomogram based on complete blood count results to predict the risk of postoperative infection in pediatric kidney transplant recipients.

## MATERIALS & METHODS

### Study population

This retrospective case-control study focused on patients under 18 years old who had received kidney transplants and were hospitalized in the Department of Pediatric Nephrology and Rheumatology at the First Affiliated Hospital of Sun Yat-sen University from January 2014 to January 2024. All patients have stable immunosuppressant drug concentration within the standard range. Exclusion criteria: (1) Transplanted kidney failure requiring hemodialysis or peritoneal dialysis again; (2) History of receiving other solid organ transplants; (3) Missing clinical data; (4) Presence of severe hematologic diseases, severe liver diseases, thyroid diseases, major trauma, or other surgeries. The data comes from the electronic medical record system and paper archived medical records of

the First Affiliated Hospital of Sun Yat-sen University. This study has been approved by the IEC for Clinical Research and Animal Trials of the First Affiliated Hospital of Sun Yat-sen University (Clinical Ethics Review, No. [2024]439). All procedures performed in this study involving human participants were in accordance with the Declaration of Helsinki (as revised in 2013). Waiver of informed consent was obtained for this study.

Pediatric kidney transplant recipients were selected and divided into two groups: the non-infected group and the infected group. The infected group of patients who developed infectious diseases after kidney transplantation. The non-infected group of patients returned to the hospital for review after kidney transplantation without any infection.

The definition of infectious diseases in renal transplant recipients adheres strictly to the diagnostic criteria set by the European Society for Clinical Microbiology and Infectious Diseases (ESCMID) (*Tacconelli et al., 2014*). Common infectious diseases among pediatric renal transplant recipients include pneumonia, upper respiratory tract infection, urinary tract infections, bacteremia, intra-abdominal infections, and skin and soft tissue infections.

## CLINICAL AND LABORATORY DATA COLLECTION

By querying the electronic medical record system, clinical data for all inpatients meeting the criteria from January 2014 to January 2024 were collected. Laboratory tests were performed on blood samples of the recipients during the follow-up period. For the infection group, data were collected at the time of hospital admission due to infection. For the non-infection group, data were gathered during routine follow-up visits, which included: (1) postoperative events requiring hospitalization, such as the removal of peritoneal dialysis catheters or long-term hemodialysis catheters, and (2) comprehensive evaluations conducted every six months following surgery.

The collected information includes the following: (1) Baseline data: gender, age, height, weight and body mass index (BMI); (2) information on infectious diseases of the kidney transplantation recipients; (3) laboratory testing indicators: white blood cell count (WBC), neutrophil count (NEUT), monocyte count (MO), lymphocyte count (LYM), platelet count (PLT), mean platelet volume (MPV), red cell distribution width-standard deviation (RDW-SD), albumin (ALB).

The following ratios were calculated: neutrophil to lymphocyte ratio (NLR), platelet to lymphocyte ratio (PLR), lymphocyte to monocyte ratio (MLR), platelet to neutrophil ratio (PNR), mean platelet volume to lymphocyte ratio (MPVLR), mean platelet volume to platelet ratio (MPR).

## STATISTICAL ANALYSIS

Statistical analyses were conducted using SPSS version 25.0 and R version 4.1.3. Normally distributed measurement data were expressed as mean ± standard deviation (M ± SD) and compared between groups using the $t$-test. Non-normally distributed measurement data were expressed as median (quartile) and compared using the Mann-Whitney $U$ test. Categorical variables were described as frequencies (percentages) and compared using chi-square tests.

Variables with $P < 0.05$ in the univariate analysis were included in the multivariate stepwise logistic regression analysis. Meaningful variables identified from the multivariate stepwise logistic regression analysis were used to construct the Nomogram prediction model. Internal validation was performed using Bootstrap resampling 1,000 times. The nomogram was evaluated using calibration curves, decision curves, and receiver operating characteristic (ROC) curves. A $P$-value $< 0.05$ was considered statistically significant.

## RESULTS

### Clinical features of the study population

A total of 305 patients who received kidney transplants were enrolled in this study. After excluding patients with a second kidney transplant ($n = 5$), peritoneal dialysis ($n = 1$), and hemodialysis ($n = 2$), 297 patients were included in the final analysis, consisting of 133 non-infected and 164 infected individuals (Fig. 1). The main types of infection were pneumonia ($n = 40$, 24.39%), urinary tract infection ($n = 25$, 15.24%), upper respiratory tract infection ($n = 32$, 19.51%), bloodstream infection ($n = 15$, 9.15%), skin and soft tissue infection ($n = 9$, 5.49%), gastrointestinal infection ($n = 21$, 12.80%), and others (*e.g.*, oral infection, abdominal infection, catheter-related infection; $n = 22$, 13.41%). Among these, viral infections ($n = 70$) were the most common, followed by bacterial ($n = 20$), mycoplasma ($n = 8$), and fungal infections ($n = 3$). No pathogens were detected in 63 children (Fig. 1).

Univariate analysis revealed statistically significant differences between the two groups for age, height, weight, WBC, NEUT, LYM, MPV, NLR, PLR, MLR, PNR, MPVLR, RDW-SD, and ALB ($P < 0.05$). However, no statistically significant differences were observed between the groups for gender, BMI, PLT, MO, and MPR ($P > 0.05$) (Table 1).

### Predictive factors identification

After incorporating the univariate significant variables ($P < 0.05$) into the regression model, the multivariate stepwise regression analysis revealed that WBC ($OR = 1.232$, 95% CI [1.090–1.392], $P = 0.001$), MLR ($OR = 19.192$, 95% CI [3.564–103.356], $P = 0.001$), PNR ($OR = 1.007$, 95% CI [1.002–1.012], $P = 0.009$), RDW-SD ($OR = 0.845$, 95% CI [0.786–0.908], $P < 0.001$), and ALB ($OR = 0.834$, 95% CI [0.776–0.896], $P < 0.001$) significantly predicted the study outcomes (Table 2).

### Validating the efficacy probability of infectious diseases using the nomogram

Based on the results of multivariate stepwise regression, we included WBC, MLR, PNR, RDW-SD, and ALB ($P < 0.05$) in constructing the nomogram. Using the five selected predictive factors, we created a column chart for the predictive model (Fig. 2). The total risk prediction score is the sum of scores from these five variables. A higher total score indicates an increased risk of the outcome, making it useful for preliminary outcome risk prediction.

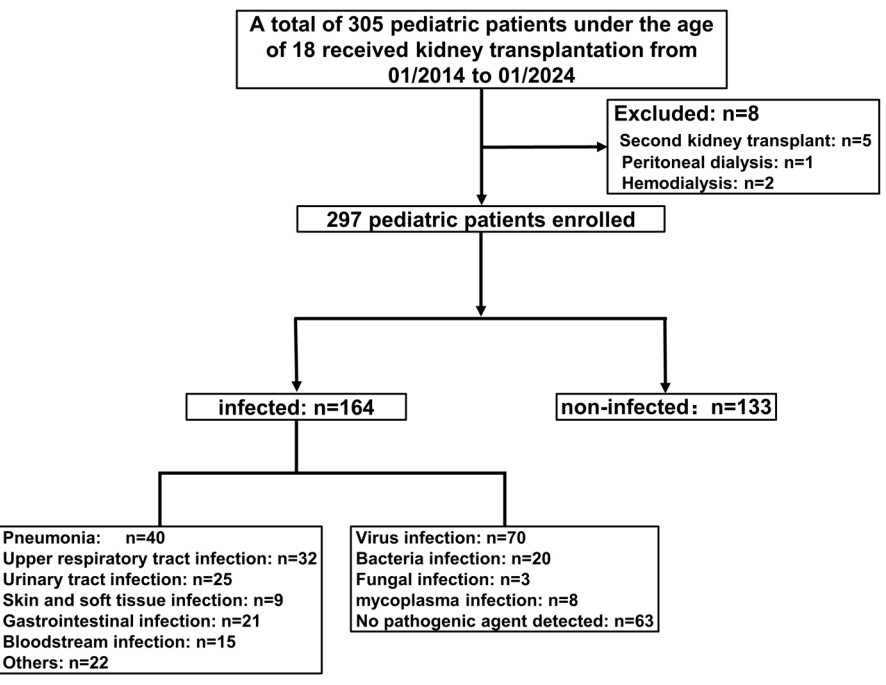

**Figure 1 Flow of participants.**

The area under the curve (AUC) for the nomogram was 0.756 (95% CI [0.702–0.811]), indicating moderate discriminatory ability, with a sensitivity of 88.0% and a specificity of 54.3% (Fig. 3).

### Calibration and decision curves for the nomogram prediction model

The internal validation using the calibration curve shows a strong concordance between the predicted and observed values of the model (Fig. 4). The calibration slope is 0.998, indicating a high level of agreement between the nomogram's predicted outcomes and the actual observations.

The clinical utility of the nomogram was evaluated using decision curve analysis (DCA). It was found that the nomogram provided increased net benefit across almost the entire threshold probability range of 10% to 90%. The decision curve indicates that the nomogram has substantial clinical utility (Fig. 5).

## DISCUSSION

The therapeutic approaches for pediatric end-stage kidney disease include hemodialysis, peritoneal dialysis, and kidney transplantation. Among these, kidney transplantation stands out as the optimal treatment modality, enhancing long-term survival rates in affected children while improving their quality of life and ensuring normal educational and daily activities (*Kalantar-Zadeh & Li, 2020*). This approach is increasingly favored by clinicians. Kidney transplant recipients require long-term immunosuppressive therapy, which, while reducing the incidence of rejection, also increases the risk of infections. The

**Table 1  Comparison of baseline characteristics and laboratory parameters between infected and non-infected groups in pediatric kidney transplant recipients.**

| Variables | Non-infected group ($n = 133$) | Infected group ($n = 164$) | Statistic | P |
|---|---|---|---|---|
| Age, M ($Q_1$, $Q_3$) | 11.00 (7.00, 13.00) | 9.00 (6.00, 12.00) | $Z = -2.22$ | 0.026 |
| Sex, $n$ (%) | | | $\chi^2 = 1.81$ | 0.178 |
|     Male | 68 (51.13) | 71 (43.29) | | |
|     Female | 65 (48.87) | 93 (56.71) | | |
| Height | 133.00 (116.00, 150.75) | 130.00 (110.00, 145.00) | $Z = -2.019$ | 0.043 |
| Weight | 30.00 (19.70, 43.85) | 26.30 (17.00, 36.50) | $Z = -2.170$ | 0.030 |
| BMI | 16.62 (14.98, 20.13) | 15.99 (14.75, 18.13) | $Z = -1.610$ | 0.107 |
| WBC, M ($Q_1$, $Q_3$) | 6.23 (5.01, 7.07) | 6.86 (4.88, 9.79) | $Z = -2.17$ | 0.030 |
| NEUT, M ($Q_1$, $Q_3$) | 3.15 (2.39, 4.02) | 3.49 (2.52, 6.43) | $Z = -2.60$ | 0.009 |
| LYM, M ($Q_1$, $Q_3$) | 2.21 (1.68, 2.88) | 1.71 (1.13, 2.76) | $Z = -3.33$ | <0.001 |
| PLT, M ($Q_1$, $Q_3$) | 278.00 (243.00, 322.00) | 275.50 (221.50, 335.00) | $Z = -0.96$ | 0.338 |
| MO, M ($Q_1$, $Q_3$) | 0.44 (0.33, 0.58) | 0.47 (0.32, 0.65) | $Z = -0.89$ | 0.376 |
| MPV, M ($Q_1$, $Q_3$) | 9.60 (8.90, 10.40) | 8.70 (8.00, 9.62) | $Z = -5.92$ | <0.001 |
| NLR, M ($Q_1$, $Q_3$) | 1.35 (1.03, 2.11) | 1.76 (0.98, 5.11) | $Z = -2.47$ | 0.014 |
| PLR, M ($Q_1$, $Q_3$) | 126.37 (97.58, 180.00) | 149.38 (95.66, 227.06) | $Z = -2.29$ | 0.022 |
| MLR, M ($Q_1$, $Q_3$) | 0.21 (0.15, 0.28) | 0.24 (0.14, 0.46) | $Z = -2.25$ | 0.024 |
| PNR, M ($Q_1$, $Q_3$) | 91.06 (68.14, 123.94) | 74.76 (46.35, 112.95) | $Z = -2.89$ | 0.004 |
| MPVLR, M ($Q_1$, $Q_3$) | 4.32 (3.36, 5.73) | 5.18 (3.40, 8.00) | $Z = -2.20$ | 0.028 |
| RDW-SD, M ($Q_1$, $Q_3$) | 40.70 (38.52, 43.28) | 39.10 (36.79, 41.02) | $Z = -3.87$ | <0.001 |
| MPR, M ($Q_1$, $Q_3$) | 0.03 (0.03, 0.04) | 0.03 (0.03, 0.04) | $Z = -1.23$ | 0.219 |
| ALB, M ($Q_1$, $Q_3$) | 41.65 (39.88, 44.80) | 39.95 (36.10, 42.35) | $Z = -5.27$ | <0.001 |

Notes.

Z, Mann-Whitney $U$ test; $\chi^2$, Chi-square test; M, Median; $Q_1$, First quartile; $Q_3$, Third quartile.

WBC, white blood cell count; NEUT, neutrophil count; MO, monocyte count; LYM, lymphocyte count; PLT, platelet count; MPV, mean platelet volume; RDW-SD, red cell distribution width-standard deviation; ALB, albumin.

NLR, neutrophil to lymphocyte ratio; PLR, platelet to lymphocyte ratio; MLR, lymphocyte to monocyte ratio; PNR, platelet to neutrophil ratio; MPVLR, mean platelet volume to lymphocyte ratio; MPR, mean platelet volume to platelet ratio.

BMI, body mass index.

**Table 2  Results of multivariate stepwise regression analysis.**

| Variables | Beta | S.E | P | OR (95% CI) |
|---|---|---|---|---|
| WBC | 0.209 | 0.062 | 0.001 | 1.232 (1.090–1.392) |
| MLR | 2.954 | 0.859 | 0.001 | 19.192 (3.564–103.356) |
| PNR | 0.007 | 0.003 | 0.009 | 1.007 (1.002–1.012) |
| RDW-SD | −0.168 | 0.037 | <0.001 | 0.845 (0.786–0.908) |
| ALB | −0.182 | 0.037 | <0.001 | 0.834 (0.776–0.896) |

Notes.

WBC, white blood cell count; MLR, lymphocyte to monocyte ratio; PNR, platelet to neutrophil ratio; RDW-SD, red cell distribution width-standard deviation; ALB, albumin.

infection risk in kidney transplant patients is 32 times higher than that in the general population (*Dharnidharka, Fiorina & Harmon, 2014*). Post-transplant patients remain on long-term immunosuppression, resulting in an increased incidence of viral, bacterial,

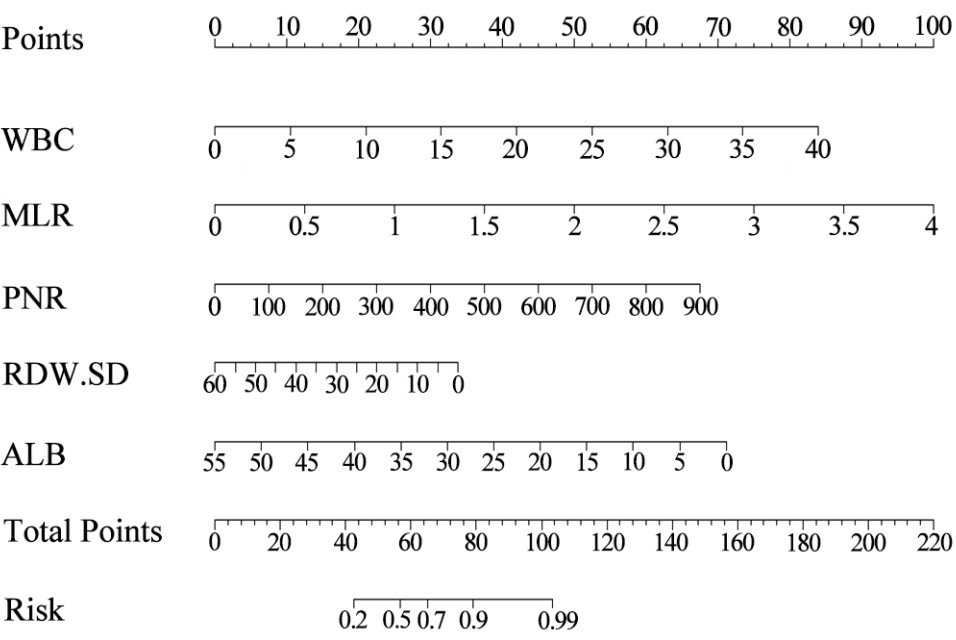

**Figure 2** **The nomogram for estimating the risk of infectious diseases in pediatric kidney transplant recipients.** Using the nomogram, the position of each variable on its respective axis corresponds to a point on the points axis. The sum of these points can then be mapped from the total points axis to the risk axis to obtain the probability of infection.

parasitic, and fungal infections (*Leal et al., 2017*), particularly during the first year following transplantation (*Vogelzang et al., 2015*).

Studies have reported that the cumulative incidence of infections in kidney transplant patients exceeds 75% during the first year (*Boan, Swaminathan & Irish, 2017*). Unlike adult recipients, young children have immature immune systems, resulting in a higher risk of infection (*Augusto et al., 2016*).

Infections have become a significant factor threatening the renal function and quality of life of kidney transplant recipients (*Kinnunen et al., 2018*; *Liu et al., 2020*). Therefore, this study aims to analyze conventional laboratory indicators and construct a predictive model to better facilitate early warning of infection risk in pediatric kidney transplant recipients. The goal is to enable proactive interventions to prevent infection-related complications.

In this study, we found that pneumonia, upper respiratory tract infection, urinary tract infection, and gastrointestinal infection were the most common types of infection, with viral diseases having the highest incidence, aligning with findings from previous studies (*Bharati et al., 2023*; *Ettenger et al., 2017*). We first performed a univariate analysis of basic clinical data (such as gender, age and BMI) and laboratory indicators, including complete blood count, albumin, and others, between the non-infected and infected groups of pediatric patients. The results indicated that age, height, weight, WBC, NEUT, LYM, MPV, NLR, PLR, MLR, PNR, MPVLR, RDW-SD, and ALB exhibited statistically significant differences between the two groups.

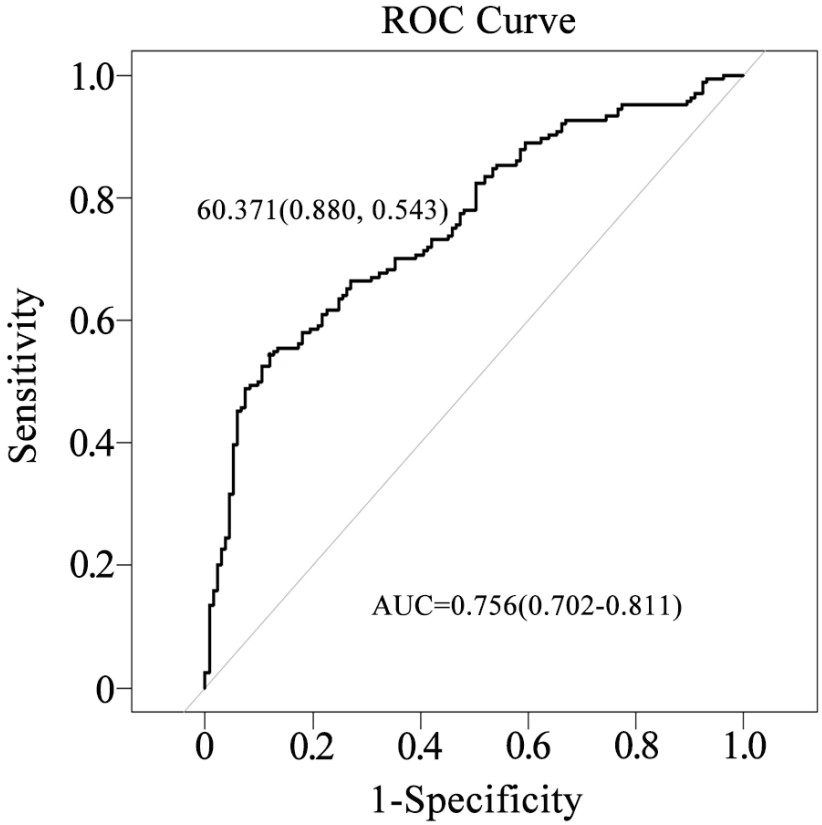

**Figure 3** **Evaluation of the nomogram's performance in predicting postoperative infection in renal transplantation using the ROC curve.** The AUC was 0.756, with a sensitivity of 88.0% and a specificity of 54.3%.

Peripheral blood cell parameters have gradually garnered increased attention. These parameters, including white blood cells, platelets, neutrophils, lymphocytes, and monocytes, can reflect the body's inflammatory response and immune status to a certain extent. However, a single count of neutrophils, lymphocytes, and platelets is easily influenced by various factors such as age. In contrast, the ratios of these cells are relatively stable and can more accurately reflect the body's inflammatory status (*Calapkulu et al., 2020*; *Larmann et al., 2020*). Peripheral blood cell count ratios, including NLR, PLR, LMR, PNR, MPVLR, and MPR, are new inflammatory markers that are easily obtainable in clinical settings. These markers have garnered increasing attention from researchers. Through univariate analysis, this study found significant differences in multiple peripheral blood cell indicators between the two groups. The statistically significant indicators from the univariate analysis were included in the multivariate stepwise logistic regression analysis. The results demonstrated that WBC, MLR, PNR, RDW-SD, and ALB had significant predictive value for the presence of infection in children with kidney transplants. RDW, as one of the blood routine indicators, reflects the degree of volume variation of red blood cells, typically increasing during inflammatory states (*Gu et al., 2022*). Moreover, RDW

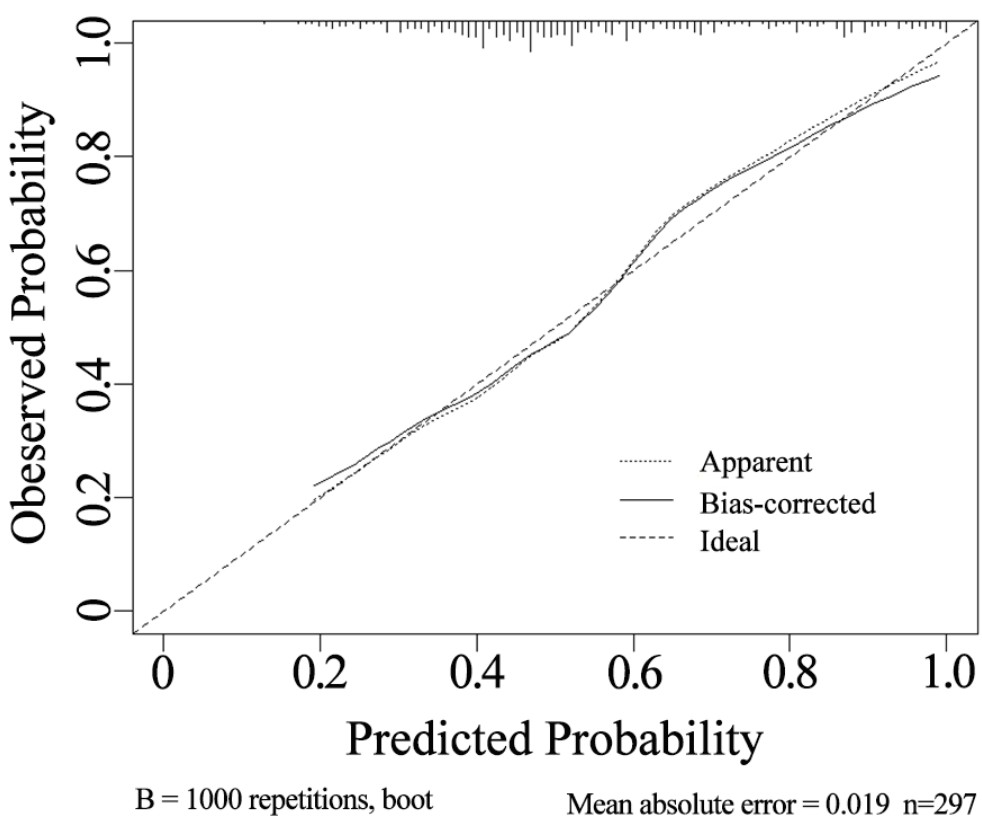

**Figure 4  Calibration curve of the nomogram.**

holds prognostic evaluation value across various diseases such as sepsis, COVID-19, and congenital heart disease (*Lee et al., 2021*; *Ramby et al., 2015*; *Silva Litao & Kamat, 2018*). The findings of this study indicate that abnormal RDW indicators may serve as predictive markers for infection in children with kidney transplants. Albumin, the most abundant plasma protein in the body, serves as a sensitive and effective biomarker that reflects the body's nutritional and inflammatory status (*Fiala et al., 2016*). Additionally, albumin plays multiple physiological roles, including anti-inflammatory, antioxidant, and anti-platelet aggregation functions (*Arques, 2018*).

Low albumin levels have been associated with an increased risk of infection (*Wiedermann, 2021*). Moreover, albumin levels tend to decrease in patients with sepsis (*Casserly et al., 2015*). Hypoalbuminemia is associated with infectious diseases following renal transplantation (*Casserly et al., 2015*). Our study also demonstrated that albumin could serve as a biomarker to predict the risk of infection in renal transplant recipients.

Nomograms are typically presented in the form of graphical representations, where variables are depicted as individual line segments, and a simplified scoring system is employed to assign scores to risk factors. This system enables the prediction of clinical event risks for individual patients (*Park, 2018*) and is now acknowledged as a reliable clinical risk prediction tool. Nomograms offer specific numerical data for the probability of event

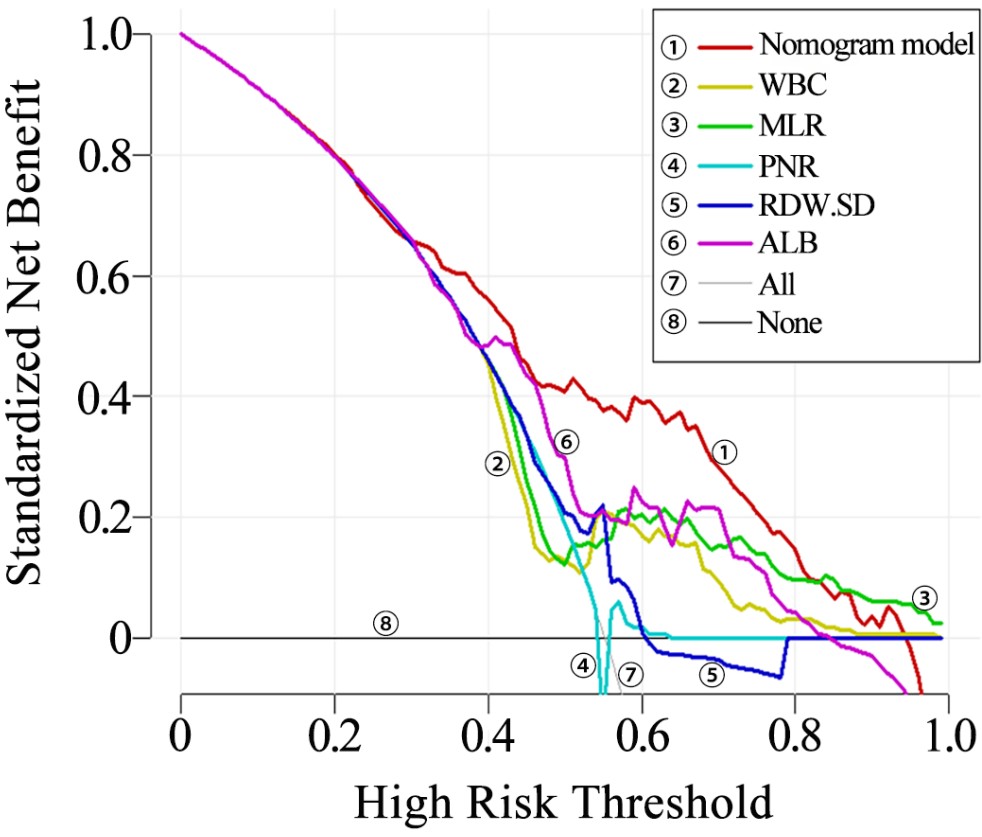

**Figure 5  Decision curve of the nomogram.**

occurrence and hold significant reference values. They have been utilized in predicting infectious diseases among adult kidney transplant recipients (*Gu et al., 2022*). Hence, our study identified 5 predictors through the results of multiple stepwise regression: WBC, MLR, PNR, RDW-SD, and ALB, and developed a nomogram. The total risk prediction score is calculated as the sum of the scores assigned to these five predictors. A higher total score indicates a greater risk of the outcome. This model can be employed for the initial prediction of infection risk in children undergoing kidney transplantation. To further assess the accuracy of the model, the Bootstrap method was employed for internal verification through 1000 resampling iterations. Subsequently, the prediction model underwent evaluation using the ROC curve. The analysis yielded an AUC of 0.756 (95% CI [0.702–0.811]), indicating strong discriminatory ability. Furthermore, calibration curves and decision curves were plotted to demonstrate good consistency between the predicted values of the model and the actual observed outcomes, highlighting its clinical practicability. These findings underscore the reliability and objectivity of the model developed in this study, which serves as an effective tool for quantifying infection risk. Moreover, it aids clinical practitioners in promptly assessing patients for infection, thereby facilitating informed clinical decisions and maximizing net benefits.

In clinical practice, CBC and albumin assessments are fundamental and routine examinations that are relatively easy to conduct, simple, and cost-effective. Rapid assessment using this model allows for the determination of infection risk scores for children, facilitating the early identification of high-risk groups vulnerable to infection. This enables precise intervention, guides treatment strategies, reduces infectious complications, and ultimately enhances the prognosis of children undergoing kidney transplantation.

In summary, WBC, MLR, PNR, RDW-SD, and ALB emerge as five effective predictors of infection in children undergoing kidney transplantation. The nomogram, constructed based on these variables, demonstrates good accuracy and clinical practicability. It facilitates early detection of infection risk in children post-kidney transplantation, enabling timely and effective preventive and therapeutic measures, and providing valuable clinical guidance.

Although the nomogram was validated and demonstrated good accuracy, there are several limitations to consider. Firstly, as this is a retrospective study only involving hospitalized children, selection bias is an inherent challenge. Secondly, the sample size in this study is relatively small. Despite using the Bootstrap method for internal validation with 1000 iterations, the study lacks validation using a large external test set and independent cohort. Future research should utilize large-scale data from research centers to further optimize the model. Additionally, different pathogens, such as bacterial, viral, or fungal, may elicit distinct responses in CBC parameters. Due to the limited positivity rates of pathogen detection, stratified analysis by pathogen type was not feasible. Future prospective studies should include larger cohorts and comprehensive pathogen testing to enhance the predictive accuracy of blood markers and develop more tailored models for managing infections in pediatric kidney transplant recipients.

### Funding
This research was funded by the Shenzhen Science and Technology Program, grant number JCYJ20220530165401004 and JCYJ20210324141008021. The funders had no role in study design, data collection and analysis, decision to publish, or preparation of the manuscript.

### Grant Disclosures
The following grant information was disclosed by the authors:
Shenzhen Science and Technology Program: JCYJ20220530165401004, JCYJ20210324141008021.

### Competing Interests
The authors declare there are no competing interests.

### Author Contributions
- Li Li conceived and designed the experiments, performed the experiments, analyzed the data, authored or reviewed drafts of the article, and approved the final draft.
- Meng Fu performed the experiments, prepared figures and/or tables, and approved the final draft.

- Changshan Wang performed the experiments, prepared figures and/or tables, and approved the final draft.
- Yuxin Pei performed the experiments, prepared figures and/or tables, and approved the final draft.
- Lizhi Chen performed the experiments, prepared figures and/or tables, and approved the final draft.
- Liping Rong performed the experiments, prepared figures and/or tables, and approved the final draft.
- Yuanyuan Xu performed the experiments, prepared figures and/or tables, and approved the final draft.
- Zhilang Lin performed the experiments, prepared figures and/or tables, and approved the final draft.
- Yuanquan Qiu performed the experiments, prepared figures and/or tables, and approved the final draft.
- Xiaoyun Jiang conceived and designed the experiments, prepared figures and/or tables, authored or reviewed drafts of the article, and approved the final draft.
- Mengjie Jiang conceived and designed the experiments, analyzed the data, authored or reviewed drafts of the article, and approved the final draft.

### Human Ethics

The following information was supplied relating to ethical approvals (*i.e.*, approving body and any reference numbers):

This study has been approved by the IEC for Clinical Research and Animal Trials of the First Affiliated Hospital of Sun Yat-sen University (No.[2024]439).

### Data Availability

Raw data are available in the Supplemental Files.

### Supplemental Information

Supplemental information for this article can be found online at http://dx.doi.org/10.7717/peerj.18454#supplemental-information.

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
