# Peer review of "Development and validation of a simple clinical nomogram for predicting infectious diseases in pediatric kidney transplantation recipients: a retrospective study"

_PeerJ, doi:10.7717/peerj.18454_

## Round 0.1 · original submission · Major Revisions

We have carefully reviewed your revisions and the feedback from our two reviewers. Both reviewers have recommended major revisions to your manuscript. After thorough consideration of their comments and your revised work, I agree with their assessment and have decided that further major revisions are necessary before we can consider your manuscript for publication. Please address all points raised by both reviewers comprehensively in your next revision. It is crucial that you pay close attention to their concerns and suggestions to substantially improve the quality and clarity of your manuscript. We look forward to receiving your revised version. Please submit your next revision along with a detailed point-by-point response to all reviewers' comments within the required timeframe.

Reviewer 1 ·

Basic reporting

The article is clearly written and conform the professional standards.

In the introduction it would be great to mention the follow-up schedule to do the CBC test.

In Study population part, there is no sample size mentioned. Also, it mentioned that the patients were selected and divided into two groups, it would be good to have a demographic comparison such as weight, height or BMI those factors will show patients baseline characteristics of the two groups.

Experimental design

In the current analysis, the authors are identifying features among WBC, NEUT, etc, which may also impact infection. While there were some key features for patients were not included such as weight, BMI, disease status, time from diagnosis to transplant, etc.

Validity of the findings

In line 134 "The total risk prediction score is the sum of scores from these five variables. A higher total score indicates an increased risk of the outcome, making it useful for preliminary outcome risk prediction."
It would be great to have a correlation justification among the five selected predictive factors.

Also, regarding the validation, the author used bootstrap method. The author may consider cross validation method to avoid overfitting.

Additional comments

The author may think about using propensity score to adjust the baseline characteristics.

Reviewer 2 ·

Basic reporting

This study aimed to develop and validate a simple nomogram for predicting the risk of infectious diseases in pediatric kidney transplant recipients. The study retrospectively analyzed clinical data from 297 hospitalized patients, identifying five key predictors of postoperative infection: white blood cell count (WBC), lymphocyte to monocyte ratio (MLR), platelet to neutrophil ratio (PNR), red cell distribution width-standard deviation (RDW-SD), and albumin (ALB). A nomogram based on these indicators showed good discriminatory ability, with an AUC of 0.756 and strong internal validation. The findings suggest this nomogram can effectively predict and evaluate the early risk of infection in pediatric kidney transplant recipients, potentially aiding in early intervention and improved patient care. Article structure, charts, raw data and references are complete. However, the writing language should be further improved to express the research purpose more clearly.

Experimental design

1) While the introduction cites some relevant studies, it could benefit from a more comprehensive review of existing research on infection prediction in kidney transplant recipients, particularly for pediatric populations. The last paragraph mentions the aim of developing a predictive model, but a more concise and specific study objective would be helpful. For example, "This study aims to develop and validate a clinical nomogram based on complete blood count results to predict the risk of postoperative infection in pediatric kidney transplant recipients."
2) The manuscript primarily focuses on the occurrence of postoperative infection in pediatric renal transplant recipients but does not clearly define infection. Furthermore, the source of the test indicators mentioned is unclear. Please include the diagnostic criteria for infection in the Methods section and provide detailed information on the timing of the collection of these test indicators.
3) How were the indicators included in this study's analysis selected, and why were creatinine, globulin, blood lipids, and blood uric acid excluded? Please suggest additional analyses or provide a thorough discussion.

Validity of the findings

1) The complete blood cell count has different responses to infection with different pathogens, and further stratified analysis or discussion should be done.
2) The manuscript states that the drug concentration of immunosuppressants in all patients was within the standard range. Please provide details on the different types of immunosuppressants used by transplant patients and discuss their impact on the risk of infection.
3) The section in Figure 3 mentions using calibration and decision curves to evaluate the nomogram. However, it doesn't provide any results from these analyses. Including information about the calibration and decision curves would enhance the understanding of the nomogram's performance and clinical utility.
4) The section in Figure 4 mentions good concordance between predicted and actual values based on the calibration curve, but it doesn't provide any specific details about the calibration curve itself. Including a more detailed description of the calibration curve, such as its shape and the magnitude of any discrepancies between predicted and observed probabilities, would be helpful. In addition, the section doesn't provide any numerical data about the calibration curve's performance, such as the Hosmer-Lemeshow statistic or the calibration slope. This information would allow readers to assess the calibration performance more objectively.
5) The section in Figure 5 states that the nomogram provides increased net benefit but doesn't provide specific details about the decision curve analysis. Including a more detailed description of the decision curve, such as the threshold probabilities where the nomogram provides a net benefit, would be helpful.

Additional comments

Future Directions: The discussion mentions future research, but it could be more specific about the types of studies needed to address the identified limitations and enhance the nomogram's clinical utility. For example, the authors could suggest future research focusing on external validation, prospective studies to assess the nomogram's impact on clinical outcomes, and investigations into the biological mechanisms underlying the identified predictors.

---

## Round 0.2 · accepted · Accept

Both reviewers have evaluated your revised manuscript and have recommended acceptance. They found that you have adequately addressed their previous concerns and improved the quality of your work. Your paper will now be forwarded to the production team for further processing.

Reviewer 1 ·

Basic reporting

The authors addressed all my previous comments. No further comments from me.

Experimental design

The authors addressed all my previous comments. No further comments from me.

Validity of the findings

The authors addressed all my previous comments. No further comments from me.

Reviewer 2 ·

Basic reporting

This manuscript has clear research objectives, uses professional English and has sufficient literature references. The experimental data are real and reliable

Experimental design

The experimental design of this manuscript meets the publication criteria, and the research results fully verify and solve the research problems proposed

Validity of the findings

The experimental design of this manuscript meets the publication criteria, and the research results fully verify and solve the research problems proposed

Additional comments

No additional comment